# Unveiling Control Vectors in Language Models with Sparse Autoencoders

## Abstract

Sparse autoencoders have recently emerged as a promising tool for explaining the internal mechanisms of large language models by disentangling complex activations into interpretable features. However, understanding the role and behavior of individual SAE features remains challenging. Prior approaches primarily focus on interpreting SAE features based on their activations or input correlations, which provide limited insight into their influence on model outputs. In this work, we investigate a specific subset of SAE features that directly control the generation behavior of LLMs. We term these "generation features", as they reliably trigger the generation of specific tokens or semantically related token groups when activated, regardless of input context. Using a systematic methodology based on causal intervention, we identify and validate these features with significantly higher precision than baseline methods. Through extensive experiments on the Gemma models, we demonstrate that generation features reveal interesting phenomena about both the LLM and SAE architectures. These findings deepen our understanding of the generative mechanisms within LLMs and highlight the potential of SAEs for controlled text generation and model interpretability. Our code is available at https://anonymous.4open.science/r/control-vector-with-sae-AAFB.

## 1 Introduction

Large Language Models (LLMs) have demonstrated remarkable capabilities in natural language understanding and generation (Brown et al., 2020; Touvron et al., 2023; Jiang et al., 2023; Bai et al., 2023). However, their black-box nature poses significant challenges, such as hallucination, bias, and factual inconsistency (Ji et al., 2023; Chang et al., 2024; Huang et al., 2023). A key reason for these challenges lies in the way individual neurons within these models encode multiple, seemingly unrelated concepts, a phenomenon known as superposition (Elhage et al., 2022). This entanglement of features complicates efforts to isolate and manipulate specific generative behaviors.

Sparse Autoencoders (SAEs) have emerged as a promising tool to address this issue by disentangling mixed representations (Bricken et al., 2023; Huben et al., 2024). SAEs map dense model activations to sparse, interpretable latent spaces, revealing latent structures that explain LLM internals. However, prior approaches mainly interpret SAE features by analyzing activations or input correlations, providing limited insights into their impact on model outputs. This gap hinders the practical utility of SAEs for precise output control.

In this work, we investigate a specific subset of SAE features, which we term **generation features**. These features act as control vectors within the LLM, reliably triggering the generation of specific tokens or semantically related token groups when activated. Notably, their influence persists across different input contexts, indicating that these features encode information that directly drives the model's generative behavior. We propose a novel causal intervention-based methodology for systematically identifying generation features. Our approach involves activating individual SAE features and measuring their causal effects on token generation probabilities to pinpoint which feature consistently controls specific outputs. Through rigorous experiments, we demonstrate that our method achieves significantly higher precision in identifying these control vectors compared to baseline approaches, such as logit lens-based methods.

Based on our method, our study reveals that generation features are concentrated in specific model regions, particularly in deeper layers, aligning with the hierarchical organization of LLMs where

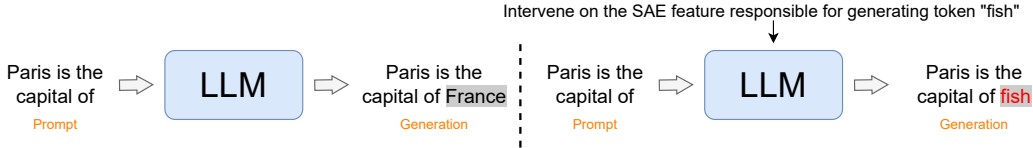

Figure 1: Illustration of generation features, which refers to specific learned features in a sparse autoencoder that contribute to the generation of certain tokens. On the left, the LLM generates the correct token "France" in response to the prompt "Paris is the capital of." On the right, an intervention is performed on the SAE feature responsible for generating the token "fish," which results in the LLM producing "fish" instead of "France." This highlights how specific features can influence the model's generation behavior in an expected way.

abstract and task-specific representations emerge. Additionally, we find that increased SAE sparsity enhances feature disentanglement and interpretability, while wider SAEs identify more features but at a lower relative density. Furthermore, by categorizing these features based on their generated outputs, we uncover patterns across token types, such as punctuation, common words, named entities, and programming-related tokens, providing insights into how LLMs organize generative knowledge through sparse, interpretable components.

**Our contribution:** (1) We introduce the concept of generation features, specific SAE features that reliably control token generation in LLMs. (2) We propose a novel causal intervention-based methodology to identify and validate these features with high precision. (3) Our analysis reveals that generation features are concentrated in specific model layers and are influenced by SAE design choices such as sparsity and width. (4) We categorize generation features based on their generated outputs, providing insights into the organization of generative knowledge in LLMs.

## 2 RELATED WORKS

**Sparse Autoencoders and Feature Disentanglement.** The phenomenon of superposition in neural networks, where individual neurons simultaneously encode multiple concepts (Elhage et al., 2022), has driven the development of sparse autoencoders (SAEs) for language model interpretation. Drawing from fundamental principles in sparse coding (Olshausen & Field, 1996), contemporary research demonstrates SAEs' capability to decompose complex representations in LLMs into interpretable features (Bricken et al., 2023; Huben et al., 2024). Bricken et al. (2023) demonstrated that stronger sparsity constraints lead to more monosemantic features, while Gao et al. (2025) introduced frameworks for scaling SAEs without compromising interpretability. Current approaches, however, predominantly analyze features through their activation patterns and input correlations, inferring feature concepts from activation circumstances (Huben et al., 2024). Consequently, the causal relationships between SAE neurons and model outputs remain insufficiently explored.

**Controllable Text Generation.** The field of controllable text generation has evolved along two primary trajectories. The first approach emphasizes decoding-time interventions, employing auxiliary networks to steer the generation process (Hu et al., 2017; Chen et al., 2019). The second operates within the latent space, beginning with efforts to learn disentangled representations during training (Hu et al., 2017; Chen et al., 2019) and progressing to recent methods for identifying and manipulating existing representations in pretrained models through steering vectors (Subramani et al., 2022; Rimsky et al., 2024). While these methods effectively control high-level generation aspects such as sentiment and topic, they primarily rely on aggregate representations or model-wide interventions. Our research advances this field by demonstrating that SAE-learned features naturally function as control vectors without requiring additional training or contrastive techniques. Moreover, our focus on individual SAE features' causal effects enables more precise control at the token and semantic concept level, leveraging the inherent disentanglement properties of these representations.

**Causal Analysis in Neural Networks.** Causal intervention frameworks (PEARL, 1995) have emerged as powerful tools for understanding information flow in transformers (Vig et al., 2020) and

mapping knowledge representations (Meng et al., 2022). Although these investigations establish foundational methods for analyzing causal relationships in neural networks, they predominantly address broad architectural components or aggregate representations. Our methodology extends these foundations by applying causal abstraction techniques specifically to SAE features, capitalizing on their disentangled nature. This synthesis enables the identification of robust causal connections between individual features and specific outputs, overcoming the traditional challenges posed by representational superposition.

## 3 PRELIMINARIES

### 3.1 SPARSE AUTOENCODER

An activation vector $a^j$ at layer $j$ can be approximated as a linear combination of feature activations and their corresponding directions:

$$a^j \approx b + \sum_i f_i(a^j)d_i, \tag{1}$$

where $b$ is a bias vector, $f_i(a^j)$ represents the activation of feature $i$, and each $d_i$ is a unit vector in activation space representing the direction of feature $i$. To learn the feature activations $f_i$ and feature directions $d_i$, we employ a sparse autoencoder. In this setup, the encoder maps the input activation $a$ to a sparse code of feature activations:

$$f(a) = \sigma(W_e a + b_e), \tag{2}$$

where $\sigma$ is a non-linear activation function, $W_e$ is the encoder weight matrix, and $b_e$ is the encoder bias. The decoder reconstructs the input activation from the sparse code:

$$\hat{a} = W_d f(a) + b_d, \tag{3}$$

where $W_d$ is the decoder weight matrix whose columns $d_i$ represent the feature directions, and $b_d$ is the decoder bias. By training the autoencoder with a sparsity constraint on $f(a)$, we encourage the model to learn a set of meaningful features $\{d_i\}$ that can effectively reconstruct $a$ from a sparse combination of feature activations.

### 3.2 CAUSAL INTERVENTION

In the context of neural networks, causal intervention involves modifying the internal activations to assess their causal impact on the model's output. Specifically, we intervene on the learned features $f(a)$ to observe how changes in feature activations affect the model's predictions. Using Pearl's $do$-operator (PEARL, 1995), we define an intervention that sets the feature activations to specific values:

$$do(f_i(a) = f_i'), \tag{4}$$

where $f_i'$ is the intervened value of feature $i$. This allows us to study the causal effect of feature $i$ on the model's output by comparing the predictions before and after the intervention.

## 4 METHODOLOGY

### 4.1 MODEL AND NOTATION

Let $M$ be a language model parametrized by a weight set $\theta$ that takes a tokenized prompt $x$ as input and returns a probability distribution over the next token for all tokens in the vocabulary $V$ of size $N_V$:

$$P_M(Y = y_i|x) = M(x;\theta)_i, \quad \forall i \in \{1, \ldots, N_V\}. \tag{5}$$

Let $a^j$ be the activation or hidden layer representation after layer $j$ within the model $M$. For clarity, we omit $j$ in the following discussion.

**Replace intervention.** A replace intervention substitutes the original activation $a$ with the scaled feature direction:

$$a' = c \cdot d_i + \epsilon. \tag{6}$$

where $d_i$ is the direction of feature $i$, $c$ indicates the strength of the intervention, and $\epsilon = N(0, 1)$ is an error term included to find robust causal effects that are resistant against random perturbations. This intervention completely replaces the activation with the feature of interest, allowing us to isolate its effect.

## 4.2 CAUSAL INTERVENTION

We perform a causal intervention on $a$ using Pearl's $do$-operator:

$$do(a = a'). \tag{7}$$

This intervention modifies the model $M$ to $M'$ with an intervened activation:

$$P_{M'}(Y|x) = P_M(Y|x, do(a)). \tag{8}$$

By comparing $P_{M'}(Y|x)$ with the original $P_M(Y|x)$, we can assess the causal effect of the intervention on the output.

## 4.3 DERIVATION FROM GENERAL CAUSAL THEOREM

Starting from the general definition of the Average Causal Effect (ACE) of generating token $y$ given feature $d$:

$$\text{ACE}(y, d) = \mathbb{E}_{x \sim D}[P_M(Y = y|x, do(a'))] - \mathbb{E}_{x \sim D}[P_M(Y = y|x)], \tag{9}$$

where $D$ is the data distribution. To make the method computationally feasible, we apply the following simplifications:

**Zero Baseline Assumption:** we assume the following property:

$$\mathbb{E}_{x \sim D}[P_M(Y = y|x)] \approx 0, \tag{10}$$

when $y$ rarely occurs without intervention. This is a valid assumption for tokens that have low prior probability in the model, which is common in large vocabularies.

**Monte Carlo Estimation:** We perform MC sampling as an estimation of the expectation.

$$\mathbb{E}_{x \sim D}[P_M(Y = y|x, do(a'))] \approx \frac{1}{N} \sum_{i=1}^{N} P_M(Y = y|x_i, do(a')), \tag{11}$$

where $N$ is the number of samples drawn from $D$. Substituting and simplifying based on our assumptions, we derive the empirical estimate:

$$\text{ACE}(y, d) \approx \frac{1}{N} \sum_{i=1}^{N} P_M(Y = y|x_i, do(a')). \tag{12}$$

For a language model that generates tokens through sampling, we estimate the probability using multiple samples:

$$\text{ACE}(y, d) \approx \frac{1}{NM} \sum_{i=1}^{N} \sum_{j=1}^{M} \mathcal{I}(\hat{y}_j = y|x_i, do(a')), \tag{13}$$

where $M$ is the number of samples per input $x_i$, $\hat{y}_{i,j}$ is the $j$-th sampled token for input $x_i$ under intervention, and $\mathcal{I}$ is the indicator function. This metric provides a practical measure to assess the causal role of the feature $d$ in generating token $y$.

## 4.4 IDENTIFYING GENERATION FEATURES

We introduce two methods for identifying generation features: the *Single-Token Analysis* and the *Multi-Token Analysis*.

### 4.4.1 Single-Token Analysis

The *Single-Token Analysis* method aims to identify features that consistently trigger the generation of a single, specific token. A feature $d$ is considered a *generation feature* of token $t$ if:

$$\text{ACE}(t, d) > \tau, \tag{14}$$

where $\tau$ is a threshold value.

**Clarification on threshold $\tau$:** We set default $\tau$ to be $0.8$ to ensure that the intervention significantly increases the likelihood of generating token $t$. Specifically, if $\text{ACE}(t, d) > 0.8$, it implies that, on average, the intervention causes the model to generate $t$ more than 80% of the time, indicating a strong causal relationship between feature $d$ and token $t$.

In practice, evaluating $\text{ACE}(y, d)$ for all tokens $y \in V$ is computationally infeasible due to the large vocabulary size. Therefore, we focus on identifying the most frequently generated token under intervention:

$$y^* = \arg\max_y \left( \frac{1}{NM} \sum_{i=1}^{N} \sum_{j=1}^{M} \mathcal{I}(\hat{y}_j = y | x_i, do(a')) \right). \tag{15}$$

We then check if $\text{ACE}(y^*, d) > \tau$ to determine if feature $d$ is a generation feature for token $y^*$.

### 4.4.2 Multi-Token Analysis

The *Multi-Token Analysis* method extends the *Single-Token method* by accounting for the possibility that a generation feature may correspond to multiple tokens, where the tokens are similar in the embedding space. Instead of looking for a single most frequent token, we analyze the generated sequences to identify a set of closely related tokens that are often triggered by the intervention.

We first obtain the generated sequences of text using the same intervention method as in the *Single-Token Analysis* method. Then, for each generated sequence, we extract the first token. For each sequence, we obtain the embedding vector for the token. These tokens are clustered using a similarity metric. Specifically, we obtain the connected components in the similarity graph formed by connecting tokens based on whether the cosine similarity between their embeddings are greater than a threshold $\theta$. We set the threshold $\theta$ to be $0.5$. Then, we define the representative set of tokens as the largest connected component found for a feature. This will output a set of tokens and their relative count given an activation. The count is the sum of the number of appearances of each token in the set.

The identified feature is considered a generation feature if the number of tokens in this set appears with the frequency that is greater than a threshold. Let $T$ be the largest cluster (set) of tokens corresponding to feature $d$, the feature is considered a generation feature if

$$\frac{1}{NM} \sum_{i=1}^{N} \sum_{j=1}^{M} \mathcal{I}(\hat{y}_j \in T | x_i, do(a')) > \tau. \tag{16}$$

In conclusion, we summarize the procedure for identifying generation features in Appendix C.

## 5 Validation of Generation Feature

### 5.1 Experimental Setup

Our experiments utilized the `google/gemma-2-2b` (Team et al., 2024) model with SAEs from Gemma Scope (Lieberum et al., 2024), employing the replace intervention method described in Section 4. We focused our analysis on layers 19 through 24, which our preliminary studies indicated contained a higher concentration of generation features. Unless otherwise specified, the SAEs used in our experiments had a width of 16,384 and an average $L_0$ norm closest to 100. For feature identification, we employed a diverse set of 10 prompts (see Appendix E), generating 10 samples per prompt to ensure robust evaluation. To assess the effectiveness of our approach, we compared our *Single-Token Analysis* method against a baseline inspired by the logit lens technique (nostalgebraist, 2020).

## 5.2 EVALUATION METRICS

Let $F$ denote the set of identified generation features, where each feature $f \in F$ has an associated target token set $t_f$ (with $|t_f| = 1$ for *Single-Token Analysis*). We define $D_{train}$ as the set of prompts used during feature identification and $D_{test}$ as an independent validation set. The specific prompts used are detailed in Appendix E. We employ two primary metrics for evaluation:

**Interventional Score.** This metric evaluates the consistency of identified generation features using an independent set of 10 validation prompts $D_{self}$. For each prompt and feature, we generate 10 samples under feature intervention with strength $c$. The interventional score for a feature $f$ is defined as:

$$\text{IS}(f) = \sum_{x \in D_{self}} \frac{\sum_{j=1}^{M} \mathcal{I}(\hat{y}_j \in t_f | x, do(a'))}{M}, \tag{17}$$

where $M = 10$ is the number of samples per input, $\hat{y}_j$ is the $j$-th sampled token under intervention $do(a')$, and $\mathcal{I}$ is the indicator function. The overall interventional score is averaged across features:

$$\text{Interventional Score} = \frac{1}{|F|} \sum_{f \in F} \text{IS}(f). \tag{18}$$

**Observational Score.** This metric assesses generalizability using activation data from Neuronpedia (Lin, 2023). It measures the likelihood that the model generates (one of) our identified target token(s) if a generation feature is naturally activated. For each feature $f$, we analyze a set of high activations $A_f$ from Neuronpedia. Each activation $a \in A_f$ corresponds to a token sequence $x_a$ and maximum activating position $p_a$. The observational score for feature $f$ is:

$$\text{OS}(f) = \frac{\sum_{a \in A_f} \mathcal{I}(t_a \in t_f)}{|A_f|}, \tag{19}$$

where $t_a$ is the token at position $p_a + 1$ in $x_a$, and $|A_f| = 5$ in our experiments. The overall observational score averages across features:

$$\text{Observational Score} = \frac{1}{|F|} \sum_{f \in F} \text{OS}(f). \tag{20}$$

**Baseline method:** Our baseline method is inspired by the logit lens (nostalgebraist, 2020). For each feature $f$, we consider its corresponding decoder weight vector $d_f$ in the sparse autoencoder. We compute the dot product between $d_f$ and the embedding vector $e_t$ for each token $t$ in the vocabulary $V$. The token $t^*$ with the highest dot product is selected as the baseline's predicted generation token:

$$t^* = \arg \max_{t \in V} (d_f \cdot e_t). \tag{21}$$

We rank features based on the highest dot product value $\max_{t \in V} (d_f \cdot e_t)$. This baseline method is directly comparable to our *Single-Token Analysis* method, as the baseline method identifies a single generation token for the generation features it finds. The precision of the baseline method is computed using the same interventional and observational procedures, substituting $\{t^*\}$ for $t_f$.

## 5.3 VALIDATION RESULTS

Tables 1 and 2 present our comprehensive validation results. Table 1 compares the performance of *Single-Token Analysis* and *Multi-Token Analysis* methods across different intervention strengths and thresholds for layers 19-24. Table 2 specifically contrasts our *Single-Token Analysis* method against the baseline approach.

**Observations:** Table 1 shows that both the *Single-Token Analysis* and *Multi-Token Analysis* methods consistently identify large numbers of generation features. The *Multi-Token Analysis* identifies more features than the *Single-Token Analysis* method. Also, both methods demonstrate a similar performance for interventional scores, with the *Multi-Token Analysis* showing a slightly lower score on observational score, but a significant increase in the number of generation features.

| Strength | Threshold | Single-Token Analysis | | | Multi-Token Analysis | | |
|---|---|---|---|---|---|---|---|
| | | # Features | Int. Score | Obs. Score | # Features | Int. Score | Obs. Score |
| 100 | 0.7 | 4308 | 0.868 | 0.609 | 6682 | 0.834 | 0.574 |
| | 0.8 | 2903 | 0.916 | 0.671 | 4276 | 0.906 | 0.651 |
| | 0.9 | 1506 | 0.953 | 0.720 | 2257 | 0.949 | 0.718 |
| 200 | 0.7 | 7748 | 0.863 | 0.537 | 11703 | 0.852 | 0.539 |
| | 0.8 | 5165 | 0.915 | 0.608 | 7930 | 0.907 | 0.604 |
| | 0.9 | 2707 | 0.953 | 0.681 | 4324 | 0.949 | 0.678 |
| 300 | 0.7 | 9554 | 0.853 | 0.496 | 14655 | 0.857 | 0.506 |
| | 0.8 | 6317 | 0.904 | 0.561 | 10013 | 0.909 | 0.572 |
| | 0.9 | 3257 | 0.948 | 0.642 | 5526 | 0.950 | 0.651 |

Table 1: Aggregated results for single-token and multi-token analysis from layer 19 to 24.

| Str. | Thres. | # Feat. | Ours | | Baseline | |
|---|---|---|---|---|---|---|
| | | | Int. | Obs. | Int. | Obs. |
| 100 | 0.7 | 4308 | 0.868 | 0.609 | 0.680 | 0.512 |
| | 0.8 | 2903 | 0.916 | 0.671 | 0.731 | 0.568 |
| | 0.9 | 1506 | 0.953 | 0.720 | 0.790 | 0.621 |
| 200 | 0.7 | 7748 | 0.863 | 0.537 | 0.530 | 0.414 |
| | 0.8 | 5165 | 0.915 | 0.608 | 0.595 | 0.478 |
| | 0.9 | 2707 | 0.953 | 0.681 | 0.683 | 0.565 |
| 300 | 0.7 | 9554 | 0.853 | 0.496 | 0.488 | 0.380 |
| | 0.8 | 6317 | 0.904 | 0.561 | 0.560 | 0.445 |
| | 0.9 | 3257 | 0.948 | 0.642 | 0.654 | 0.534 |

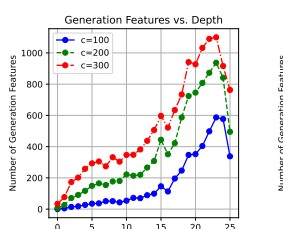 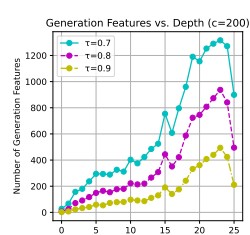

(a) Effect of intervention strength $c$; Effect of depth   (b) Effect of threshold $\tau$; Effect of depth

Table 2: Comparison of single-token analysis and baseline. The results are from layers 19 through 24. Our method consistently achieves higher interventional and observational scores.

Figure 2: Layer-wise distribution of generation features. (a) Variations with different intervention strengths (fixed $\tau = 0.8$); (b) Variations with different thresholds (fixed $c = 200$).

Table 2 demonstrates that our *Single-Token Analysis* method achieves significantly higher interventional and observational scores than the baseline method across all intervention strengths and thresholds, demonstrating the superiority of our causal intervention-based approach for identifying generative neurons compared to a logit lens baseline.

# 6 GENERATION FEATURES STUDY

## 6.1 LOCATION OF GENERATION FEATURES

We analyzed the distribution of generation features across different model layers to understand their formation patterns, motivated by recent work on layer specialization (Jin et al., 2025) and layer functionality (Gromov et al., 2025; Zhang et al., 2024). Using the *Single-Token Analysis* method, we examined feature distributions across layers in the gemma-2-2b model.

Figure 2 illustrates the layer-wise distribution under varying experimental conditions. Both intervention strength and threshold variations reveal a consistent pattern: generation features become increasingly prevalent in deeper layers, reaching peak concentration in the later layers before showing a slight decline in the final layer. This pattern persists across different parameter settings, suggesting a fundamental aspect of how these models organize generative capabilities.

The increasing density of generation features in later layers aligns with the hierarchical nature of transformer architectures, where deeper layers typically process more abstract and task-specific features. The slight decrease in the final layer may indicate a transition to output-specific processing, where individual feature effects become more diffused.

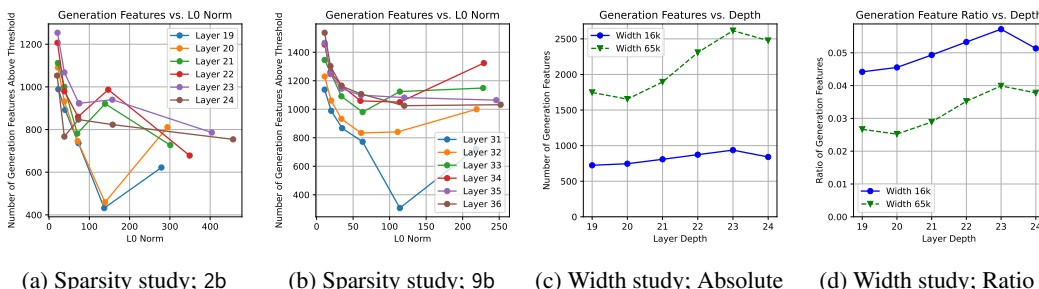

(a) Sparsity study; 2b    (b) Sparsity study; 9b    (c) Width study; Absolute    (d) Width study; Ratio

Figure 3: Impact of SAE sparsity and width. (a) and (b) show generation feature count versus SAE $L_0$ norm for `gemma-2-2b` and `gemma-2-9b`, respectively. (c) and (d) compare generation features across different SAE widths (absolute count of generation features and ratio of generation features).

## 6.2 IMPACT OF SAE SPARSITY

We investigated the relationship between SAE sparsity and generation feature formation through an ablation study varying the average $L_0$ norm. The average $L_0$ norm, representing the average number of non-zero activations in the SAE's hidden layer, directly controls the sparsity level of the learned representations. Prior research suggests that sparsity levels influence feature interpretability and reconstruction quality (Bricken et al., 2023; Chanin et al., 2024), with higher sparsity often yielding more interpretable features despite potential increases in reconstruction loss.

Our analysis focused on layers 19-24 of `gemma-2-2b` and layers 31-36 of `gemma-2-9b`, using SAEs with width 16k, intervention strength $c = 200$, *Single-Token Analysis* and threshold $\tau = 0.8$. As shown in Figure 3a and 3b, for average $L_0$ norms below 100, we observe a general trend where lower average $L_0$ norms (higher sparsity) correspond to more identified generation features. This relationship becomes less clear beyond average $L_0$ norm of 100, where the feature count shows some variability and occasional increases, likely due to the complex interplay between sparsity and feature representation.

The observed pattern in the low average $L_0$ norm region may be attributed to the feature disentanglement effect of high sparsity, where features are forced to be more distinctive and specialized. In contrast, the variable behavior at higher average $L_0$ norms suggests that reduced sparsity constraints allow for more complex feature interactions, potentially leading to both feature splitting (a single concept starts to be represented by multiple features) and merging (multiple concepts become encoded in a single feature) phenomena discussed in (Chanin et al., 2024).

## 6.3 IMPACT OF SAE WIDTH

We examined how SAE width influences generation feature formation by comparing results across layers 19-24 in `gemma-2-2b` for widths of 16k and 65k. Figure 3c and 3d presents this comparison using a fixed intervention strength of $c = 200$ with *Single-Token Analysis* and threshold $\tau = 0.8$.

While the wider 65k SAE identifies more generation features in absolute terms, the ratio of generation features to total width is actually lower compared to the 16k SAE. This suggests that simply increasing SAE width does not proportionally increase the density of generation features. We leave the study of the scaling and explanation of this phenomenon to future research.

## 6.4 CHARACTERISTICS OF GENERATION FEATURES

To better understand the nature of identified generation features, we conducted a comprehensive analysis of their distribution and characteristics. Using LLM, we categorized all 1,202 unique generation tokens (corresponding to 10,136 features) into six distinct categories. Table 3 presents this categorization along with representative examples selected from the most frequent tokens within each category, as detailed in Appendix D.2. We also provide the implementation of the categorization in Appendix D.1.

| Category | #Tokens | #Features | Example Tokens |
|---|---|---|---|
| Punctuation & Symbols | 92 | 3,794 | ".", ",", "(", "{", "-" |
| Common Words & Function Words | 189 | 3,261 | "of", "to", "the", "in", "and" |
| Numbers & Digits | 15 | 256 | "0", "1", "2", "4", "3" |
| Proper Nouns & Named Entities | 52 | 136 | "al", "R", "University", "arXiv", "com" |
| Programming & Code-Related | 185 | 737 | "://", "<tr>", "www", "x", "return" |
| Content Words | 669 | 1,952 | "item", "all", "much", "get", "about" |

Table 3: Distribution of generation features across categories. Example tokens are selected from the most frequent tokens in each category, as detailed in Appendix D.

| Prompt | Feature (Layer, ID) | Original Continuation | Intervention Result |
|---|---|---|---|
| The weather today seems unusually bright and | Layer 22, ID 9836 | sunny. I'm not sure if it's because of the time of year... | black. The air is clear as a clear day... |
| The 44th president of USA is | Layer 25, ID 2222 | a man who has been in the limelight for a long time... | Trump. The 44th President is the president of USA... |
| 1+1= | Layer 7, ID 1139 | 2... | 112 - 113 (20 points)... |

Table 4: Examples of generation feature interventions. Target tokens to generate are "black", "trump", and "1" from top to bottom. Intervention on one feature at one layer for one token effectively changes the model behavior.

### 6.5 EXAMPLES OF GENERATION FEATURE INTERVENTION

To demonstrate the practical impact of generation features, we present several examples of how targeted interventions can alter model outputs. Table 4 shows three representative cases where activating specific generation features leads to consistent changes in the model's continuation.

In each case, we observe that activating a specific generation feature (using replace intervention with strength 200) consistently redirects the model's output toward a particular token or concept, regardless of the contextual appropriateness. For instance, in the weather example, activating feature 9836 in layer 22 consistently generates "black" instead of the more contextually appropriate "sunny". This demonstrates how generation features can override context-based generation patterns.

## 7 CONCLUSION

In this work, we introduced a novel methodology for identifying and validating generation features—specific sparse autoencoder (SAE) features that reliably control token generation in large language models (LLMs). By systematically applying causal interventions on SAE activations, we demonstrated that these features act as control vectors, consistently influencing the generation of specific tokens or token groups across diverse contexts. Our experiments on the Gemma models revealed that generation features are concentrated in deeper layers, aligning with the hierarchical organization of transformer architectures. We further explored how SAE architectural choices, such as width and sparsity, impact feature interpretability and density, finding that higher sparsity enhances disentanglement while larger widths increase absolute feature counts but lower relative density. Qualitative examples showcased the practical implications of generation features, illustrating their capacity to override contextual outputs and directly control model behavior. These findings provide new insights into the internal organization of LLMs, offering a systematic framework for both understanding and precisely controlling their generative capabilities. While these findings offer promising directions for controlling LLM behavior, they also raise important ethical considerations (see Appendix B). Future work can extend this approach to larger models and explore its applications for fine-grained, interpretable interventions in LLMs while maintaining output coherence and safety.

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

## A    LIMITATIONS

Our analysis primarily focuses on the Gemma 2 2B model using the Gemma Scope SAE suite (Lieberum et al., 2024), with preliminary experiments on Gemma 2 9B. The generalization of our findings to other model architectures (e.g., LLaMA (Touvron et al., 2023), Mistral (Jiang et al., 2023)) or SAE architecture beyond JumpReLU (Rajamanoharan et al., 2024) used in Gemma Scope remains to be verified.

## B    POTENTIAL RISKS

The ability to precisely control LLM outputs through generation features, while valuable for research and legitimate applications, carries several potential risks:

- **Adversarial Manipulation**: Generation features could be exploited to override model safeguards or inject unwanted content into model outputs.

- **Bias Amplification**: Targeted activation of certain features might amplify existing biases or introduce new ones into model responses.

- **Misuse in Misinformation**: This technique could be used to force models to generate specific narratives, potentially facilitating the spread of misinformation.

We encourage researchers to carefully consider these risks when building upon this work and to implement appropriate safeguards in practical applications.

## C    ALGORITHM DETAILS

---

**Algorithm 1** Algorithm for Identifying Generation Features (Replace Intervention)

---

**Require:** Model $M$, dataset $D$, feature directions $\{d_i\}$, intervention strength $c$, threshold $\tau$
**Ensure:** Set of generation features $G$

1: **Initialize:** $G \leftarrow \emptyset$
2: **for** each feature $d_i \in \{d_i\}$ **do**          ▷ Iterate through all feature directions.
3:     **for** each sample $x \in D$ **do**          ▷ Iterate through all samples in the dataset.
4:         $a \leftarrow \text{Activation}(M, x)$          ▷ Compute the original activation.
5:         $\epsilon \leftarrow \text{SampleNoise}()$          ▷ Sample noise from $N(0, 1)$.
6:         $a' \leftarrow c \cdot d_i + \epsilon$          ▷ Apply the replace intervention.
7:         **for** $j = 1$ to $M$ **do**          ▷ Sample $M$ tokens from the model.
8:             Sample token $\hat{y}_{x,j} \sim P_M(Y \mid x, \text{do}(a'))$
9:         **end for**          ▷ Record the frequency of each token generated.
10:     **end for**
11:     **Single-Token Analysis:** ▷ Estimate $\text{ACE}(y, d_i)$ for each token $y$ using the collected samples.
12:     $y^* \leftarrow \arg\max_y (\text{ACE}(y, d_i))$          ▷ Find token with maximum ACE.
13:     **if** $\text{ACE}(y^*, d_i) > \tau$ **then**
14:         Add $(d_i, y^*)$ to $G$
15:     **end if**
16:     **Multi-Token Analysis:** ▷ Cluster generated tokens based on embedding similarities to find set $T$.
17:     **if** $\frac{1}{NM} \sum_{x \in D} \sum_{j=1}^{M} \mathcal{I}(\hat{y}_{x,j} \in T \mid \text{do}(a')) > \tau$ **then**
18:         Add $(d_i, T)$ to $G$
19:     **end if**
20: **end for**
21: **return** $G$

---

# D DETAILED ANALYSIS OF GENERATION FEATURES

## D.1 FEATURE CATEGORIZATION METHODOLOGY

We employed the DeepSeek-V2.5 model to categorize generation features using a systematic prompt-based approach. The categorization prompt was structured as follows:

```
Please categorize the following tokens
into one of these categories:
1. Punctuation and Symbols
2. Common Words and Function Words
3. Numbers and Digits
4. Proper Nouns and Named Entities
5. Programming and Code-Related Tokens
6. Content Words

Example categorization:
Input tokens: ["!", "and", "1",
"John", "class", "book"]
Output:
!: 1
and: 2
1: 3
John: 4
class: 5
book: 6
```

## D.2 TOP FEATURES BY CATEGORY

| Token | Feature Count |
|---|---|
| . | 1,155 |
| , | 510 |
| ( | 370 |
| { | 240 |
| - | 227 |
| " | 109 |
| / | 94 |
| ; | 88 |
| = | 83 |
| _ | 73 |

Table 5: Top Punctuation & Symbols

| Token | Feature Count |
|---|---|
| of | 626 |
| to | 488 |
| the | 219 |
| in | 129 |
| and | 109 |
| for | 98 |
| as | 88 |
| on | 71 |
| with | 68 |
| from | 62 |

Table 6: Top Common Words & Function Words

| Token | Feature Count |
|---|---|
| 0 | 103 |
| 1 | 59 |
| 2 | 49 |
| 4 | 9 |
| 3 | 7 |
| 9 | 7 |
| 5 | 6 |
| 20 | 4 |
| 6 | 2 |
| 7 | 2 |

Table 7: Top Numbers & Digits

| Token | Feature Count |
|---|---|
| al | 14 |
| R | 9 |
| University | 8 |
| arXiv | 7 |
| com | 7 |
| office | 7 |
| City | 5 |
| Dr | 5 |
| God | 4 |
| Microsoft | 4 |

Table 8: Top Proper Nouns & Named Entities

| Token | Feature Count |
|---|---|
| :// | 48 |
| <tr> | 27 |
| www | 24 |
| x | 24 |
| return | 21 |
| <tbody> | 19 |
| <td> | 19 |
| class | 18 |
| function | 15 |
| php | 15 |

Table 9: Top Programming & Code-Related Tokens

| Token | Feature Count |
|---|---|
| item | 24 |
| all | 23 |
| much | 21 |
| get | 20 |
| about | 18 |
| new | 17 |
| \ | 16 |
| true | 16 |
| public | 14 |
| said | 13 |

Table 10: Top Content Words

# E    PROMPTS USED FOR EXPERIMENTS

## E.1    PROMPTS FOR IDENTIFICATION

```
prompts:
- "<unk>␣<unk>␣<unk>␣<unk>"
- "He␣finally␣realized␣his"
- "The␣ancient␣library␣held"
- "Whispers␣echoed␣in␣the"
- "They␣raced␣against␣the"
- "Remembering␣the␣days␣when"
- "If␣only␣she␣had␣known"
- "In␣the␣future␣we␣will"
- "The␣door␣creaked␣open,␣revealing"
- "In␣the␣land␣of␣make-believe"
```

## E.2    PROMPTS FOR VALIDATION

```
prompts:
- "The␣weather␣today␣seems␣unusually␣bright␣and"
- "She␣quickly␣realized␣that␣her␣favorite␣book␣was"
- "By␣the␣time␣the␣concert␣ended,␣the␣crowd"
- "The␣scientist's␣discovery␣led␣to␣a␣groundbreaking"
- "While␣hiking␣through␣the␣forest,␣I␣stumbled␣upon"
- "Despite␣the␣warnings,␣he␣decided␣to"
- "The␣software␣update␣introduced␣several␣new␣features␣that"
- "After␣years␣of␣research,␣the␣team␣concluded␣that"
- "As␣the␣plane␣ascended,␣the␣passengers␣could␣see"
- "He␣always␣wondered␣why␣the␣stars␣seemed␣to"
```

