# OpenReview forum: "Unveiling Control Vectors in Language Models with Sparse Autoencoders"
_ICLR.cc/2025/Workshop/BuildingTrust — Submitted to BuildingTrust_

### Official Review · Reviewer_B3KQ · 2025-02-16

**Rating:** 4
**Confidence:** 4

**Review:**

Strengths

* **Clear Methodology** - Paper presents a systematic approach based on causal intervention for identifying generation features. The methodology is explained thoroughly and builds logically from established theoretical foundations (eg do operator).

* **Well-Structured Organization** - The paper maintains structural coherence throughout and was easy to follow.

* **Somewhat Comprehensive Empirical Analysis** - The research provides empirical investigation of Gemma models, specifically examining how sparsity and width influence generation features. Their layer-wise distribution analysis offers some valuable insights into model behavior and feature interactions. However, I would have liked to see models from outside the Gemma family been evaluated as well, given the availability of open-weights SAEs (eg Llama-Scope)

Weaknesses

* **Limited Innovation and Practical Impact** - The fundamental concept of identifying SAE features that predictably increase logits for specific token sets lacks novelty within the field. Similar approaches have been extensively explored in interpretability research, and the paper doesn't sufficiently differentiate its contributions. The single-token focus appears arbitrary and lacks clear practical value. The paper's central concept of "generation features" appears somewhat self-evident, as the existence of correlations between sparse features and token generation could be reasonably expected. The research fails to demonstrate why this finding represents a significant advancement in our understanding of language models.

* **Overly Restricted Scope** - The research's emphasis on single-token or small token set control significantly limits its applicability. While the authors acknowledge broader work on style and sentiment control, they don't adequately justify their narrow focus. This limitation substantially reduces the practical utility of their findings for real-world applications. Despite mentioning possible applications in controlled text generation and model interpretability, the paper fails to demonstrate compelling real-world use cases.

* **Inadequate Theoretical Foundation** - The "Zero Baseline Assumption," while convenient for analysis, lacks robust justification. The paper would benefit from a more thorough theoretical explanation and empirical validation of this assumption, particularly regarding its applicability across different contexts.

Questions
* See Weaknesses

---

### Official Review · Reviewer_zugh · 2025-02-19
**Interesting application of SAEs to controllable generation; weak accept**

**Rating:** 6
**Confidence:** 3

**Review:**

SAEs are used to find the building block features of model activations. There are two problems with this: 1) because this is an unsupervised method, we have to manually identify the semantic meaning of SAE latents which is difficult 2) there is no guarantee that latents are causally used by the model. This paper identifies a subset of latents that have causal power in making the model output specific generations when steered with. Steering here is setting activations to certain values.

Overall, I find the use of SAEs for controllable generation to be an interesting application. I weakly accept because I believe better comparative baselines could have been chosen.

Strengths
- The paper identifies an interesting two-way question. 1) Can controllable generation be used to interpret SAE latents and 2) can SAE latents be used to help controllable generation. Both are interesting directions. The latter especially is useful, because mechanistic interpretability as a field is struggling to find real-world use cases for SAEs.
- The paper is well written and has many relevant experimental results. There are good robustness checks on SAE width and sparsity. I also liked the investigation of layer depth.

Weaknesses
- It's not clear to me how useful controllable generation is, especially in the single-word regime. Multi-token controllable generation seems more useful, but I fail to see the utility of having a model that predictably always outputs a single phrase. It's just an if-statement at that point.
- I think this paper could be made stronger by comparing against other non-SAE baseline method. I.e., the exploration of the logit lens baseline still uses SAE features as a basis to consider. What about a supervised baseline where we take a hundred prompts where the next token is "black," take the average activation on the last token, and use that to steer with rather than the SAE latent for "black" (Table 4 first row). There's an argument to be made that unsupervised methods are more valuable than supervised probes, but I think additional discussion that considers stronger baselines could improve the paper.

---

### Official Review · Reviewer_uyKU · 2025-03-02
**Identifying SAE latents with a large effect on language model outputs**

**Rating:** 2
**Confidence:** 5

**Review:**

This paper demonstrates an approach for understanding which SAE latents have a large effect on a language model’s outputs. The paper is hard to follow and lacks detailed explanations for important assumptions. It doesn’t provide a clear motivation for why the its perspective of “generation latents” is a more useful model then measuring the attribution of a latent on outputs.

# Strengths
- The paper touches on an interesting direction. Identifying SAE latents with a large effect on the output could produce useful, practical applications of interpretability.

# Weaknesses
- The formalization of causal interventions with do-calculus is unnecessary, lengthy, and detracts from the narrative of the paper.
- Generation is measured by how often scaling the latent leads to a specific token appearing in the model’s outputs. There is no analysis done on whether steering the latent scales the log-probs of the token which might provide a cheaper, more granular analysis of generation latents.
- “Generation latents” is an inaccurate perspective. Scaling a latent has some downstream effect, mediated by intermediate components in the transformer. Some latents have a more direct effect on the outputs than others.
- There’s no motivation for measuring the correlation between l0 and the number of generations latents per layer.
- Categories are chosen arbitrarily without an explanation for why they are important types of generation latents to understand.
- Ten prompts for latent identification is not enough to make a significant claim on the number of SAE latents.
- The amount that a latent needs to be scaled varies; there’s no information on the scaling coefficients for specific latents. Scaling equally for all latents might not elicit information.
- There’s no explanation on what the error term in equation 6 is for.

---

### Decision · Program_Chairs · 2025-03-04

**Decision:**

Reject

**Comment:**

The paper lacks a clear motivation for its "generation latents" perspective and does not convincingly demonstrate its novelty beyond existing interpretability research. Additionally, its scope is overly restricted to single-token control, limiting real-world applicability, while key methodological choices—such as causal intervention formalization and latent scaling—lack sufficient justification or comparative baselines.